# Targeting the Structural Maturation Pathway of HIV-1 Reverse Transcriptase

**DOI:** 10.3390/biom13111603

**Published:** 2023-11-01

**Authors:** Thomas W. Kirby, Scott A. Gabel, Eugene F. DeRose, Lalith Perera, Juno M. Krahn, Lars C. Pedersen, Robert E. London

**Affiliations:** Genome Integrity and Structural Biology Laboratory, National Institute of Environmental Health Sciences, NIH, Research Triangle Park, Durham, NC 27709, USAjuno.krahn@nih.gov (J.M.K.);

**Keywords:** HIV-1 reverse transcriptase, RT structural maturation, maturation inhibitors, RT polymerase domain, ground state stabilization, RT dimerization inhibitor

## Abstract

Formation of active HIV-1 reverse transcriptase (RT) proceeds via a structural maturation process that involves subdomain rearrangements and formation of an asymmetric p66/p66′ homodimer. These studies were undertaken to evaluate whether the information about this maturation process can be used to identify small molecule ligands that retard or interfere with the steps involved. We utilized the isolated polymerase domain, p51, rather than p66, since the initial subdomain rearrangements are largely limited to this domain. Target sites at subdomain interfaces were identified and computational analysis used to obtain an initial set of ligands for screening. Chromatographic evaluations of the p51 homodimer/monomer ratio support the feasibility of this approach. Ligands that bind near the interfaces and a ligand that binds directly to a region of the fingers subdomain involved in subunit interface formation were identified, and the interactions were further characterized by NMR spectroscopy and X-ray crystallography. Although these ligands were found to reduce dimer formation, further efforts will be required to obtain ligands with higher binding affinity. In contrast with previous ligand identification studies performed on the RT heterodimer, subunit interface surfaces are solvent-accessible in the p51 and p66 monomers, making these constructs preferable for identification of ligands that directly interfere with dimerization.

## 1. Introduction

HIV-1 reverse transcriptase was the first HIV enzyme to be successfully targeted with a therapeutic drug and its inhibition remains a cornerstone of HIV treatment [1,2,3]. Two types of inhibitors have proven successful for clinical applications: nucleoside reverse transcriptase inhibitors (NRTIs) and non-nucleoside reverse transcriptase inhibitors (NNRTIs) [4,5,6,7]. NRTIs are typically intracellularly triphosphorylated and bind at the active enzyme site where they may be incorporated into the primer terminus, while NNRTIs interact with a dynamically accessible site that is not present in structures of the uncomplexed enzyme [8,9]. Inhibitors that target the separate active site in the RT RNase H domain have also been developed but, at present, have not proven sufficiently efficacious to progress to clinical use [10,11]. The viral RNA codes for a p66 protein that forms an asymmetric homodimer in which the RNase H domain in one subunit becomes destabilized [12], leading to subsequent unfolding and proteolysis in order to form the active p66/p51 heterodimer [13,14,15]. An analogous process in HIV-2 creates a p68/p58 heterodimer [14,16]. Conversion of the p66 monomer to the heterodimer involves a series of steps that include a metamorphic transition, required to convert the inactive monomer structure into the active structure that is present in the mature enzyme. Existing drug therapies have proven highly useful but are limited by toxicity—particularly the mitochondrial toxicity of NRTIs [17], as well as by the remarkable ability of the enzyme to develop drug-resistant variants [4,5,6]. For these reasons, work continues on the development of new strategies for RT inhibition [18,19].

The complex structural maturation process required to form the RT heterodimer represents another potential drug target. We previously had found that the final step in this process, the subunit-specific unfolding of one of the RNase H domains, can be slowed approximately fivefold using a ligand that binds to the RNase H active site and stabilizes the RNase H domain [12]. This unfolding step is essential in order to make the F440|Y441 proteolytic cleavage site accessible to HIV-1 protease. However, it is unclear whether such a reduced processing rate would be sufficient to significantly interfere with the viral life cycle, since the activity of the p66/p66′ homodimer is nearly as high as that of the p66/p51 heterodimer [20,21].

In the present study, we have evaluated the possibility of interfering with the initial steps of this process—the structural subdomain rearrangement of the monomer and subsequent dimerization. The p66 monomer represents an attractive target for drug development for several reasons. First, targeting the process at the initial isomerization step is more likely to be effective in blocking the complete maturation pathway. Second, the p66 monomer structure has been determined [13] and shown to include a compact, globular fingers/palm/connection (F/P/C) domain core that is very similar to that of the p51 subunit in the mature RT heterodimer, with the thumb and RNase H domains linked by flexible segments. The globular F/P/C structure present in both the p51 and p66 monomers as well as in the p51 subunit of RT (Figure 1) provides a conveniently available structure for inhibitor development. Third, the ligand screening studies of Arnold and colleagues [22,23,24] are useful for the identification of binding cavities located at subdomain interfaces. These provide a basis for the further identification of higher affinity ligands by computational screening. Fourth, the dimer interface that is buried in the RT homo- or heterodimer is solvent-exposed in both the p51 and p66 monomers as well as in the F/P/C constructs used in this study, allowing direct detection of ligand interactions with surface interfaces that are buried and, therefore, experimentally inaccessible in the dimer. In this study, we have evaluated several initial candidates identified using computer-aided analysis of the p51 binding sites.

## 2. Materials and Methods

### 2.1. Chemicals Evaluated

With one exception, the chemicals evaluated in this study were obtained either from commercial sources or, in quantities less than 5 mg, from the repository of synthetic and pure natural products maintained by the developmental therapeutics program of the National Cancer Institute. Picric acid was obtained from Sigma-Aldrich (St. Louis, MO, USA) and neutralized with sodium hydroxide to yield a sodium picrate solution prior to use. The 9H-Xanthene-1,3,6,8-tetrol (XOH) was obtained from Omicron Biochemicals, Inc. (South Bend, IN, USA) as a custom synthesis.

### 2.2. Protein Design and Expression

Fingers/palm/connection (FPC) constructs were designed as described above with deletions of the disordered thumb and the C-terminal residues, as well as an additional deletion of the palm C-terminal segment in order to block the ability of the p51 (or p66) ground state to adopt the rearranged excited state structures. In these studies, we used either the same deletion (Lys219–Met230) described in our previous studies [13] or an alternate deletion in the same region (Thr215–Lys223) corresponding to the disordered residues in chain B of structure PDB: 5CYQ [22]. Because deletion of the thumb creates a significant gap of almost 19 Å between the terminal palm Thr^240^ and connection Tyr^319^ residues, an additional segment of sequence GSGSGG was added to connect these two residues. Swiss model was used to make sure that substitution of the connecting sequence for the thumb subdomain did not significantly alter the structure of the remaining FPC subdomains. Met357 is a poorly conserved, surface-exposed residue that produces an intense resonance that tends to obscure observation of more useful resonances. Consequently, FPC1 and FPC2 were designed to include M357R mutations. The sequences used are given in Appendix A. The FPC1 and FPC2 sequences in a pET40 expression vector were obtained from Genscript (Piscataway, NJ, USA).

### 2.3. Expression of FPC

Selective labeling of large proteins with [^13^CH_3_-methionine] provides a useful approach for obtaining structural, dynamic, and binding information near the labeled sites. However, due to the large size of the RT heterodimer, the observation of methionine resonances is limited to those with greater solvent exposure [25,26]. FPC1 and FPC2 plasmids and the variant plasmids were used to transform *E. coli* strain BL21(DE3); then, the unlabeled proteins were expressed by autoinduction [27] using ZYM-5052 media with 100 µg/mL kanamycin. [^13^CH_3_-Met]-labeling was affected by autoinduction using MDA-5052 media with 100 µg/mL kanamycin. The autoexpressed FPC proteins were purified by IMAC chromatography using a 5 mL HisTrap column (GE Healthcare, Chicago, IL, USA), followed by gel filtration chromatography using a Superdex 75 column (GE Healthcare, Chicago, IL, USA). Purified proteins were concentrated and stored at −70 °C.

### 2.4. Expression of p51

A pET30-based expression vector for p51 protein (with the C38V/C280S/M357T substitutions) was obtained from Genscript and the protein was autoinduced using ZYM-5052 media with 100 µg/mL kanamycin. For the p51 dimerization assay, it was necessary to obtain p51 without an affinity tag. This requirement arises because any divalent metal ions in the preparation can bind to one or two affinity tags, leading to metal ion-mediated dimer formation. Indeed, we were able to observe this effect in studies of His-tag-p51 constructs in the presence of multiple metal ions, including Zn^2+^ and Mn^2+^ (unpublished results). This construct has no affinity tag, so the protein was purified by affinity to a HiTrap heparin (GE Healthcare) column and gel filtration chromatography using a Superdex 75 column.

### 2.5. Expression of FPC Variants for Resonance Assignments

In previous studies of [^13^CH_3_-Met]RT, many of the methionine methyl resonances in the p51 subunit were not observed due to broadening. Exceptions included Met18 as well as Met230, which is located in a disordered region of the structure (see above) and gives a relatively intense peak. Alternatively, the much smaller size of the F/P/C constructs (~40 kD) reduces the broadening contribution of overall molecular rotational tumbling, so that previously unobserved resonances corresponding to residues Met41 and Met164 could be observed in the F/P/C constructs. The Quik-Change site-directed mutagenesis kit (Agilent) was used to create an expression vector for the FPC1 M→L mutations corresponding to each of the five Met residues: M16, M41, M164, M184, and M230. Expression and purification of the variant proteins were the same as for the wild type. We also investigated several Quik-Change-generated mutations, including L234M and E203S. The first was used to introduce an NMR reporter Met residue closer to the BYZ4 site and the second in an unsuccessful effort to improve crystallization behavior of these constructs.

### 2.6. NMR Studies

#### 2.6.1. STD Experiments

STD studies [28] were performed on samples containing 15 µM FPC constructs 1 or 2, 200 µM ligand in the following buffer: 50 mM potassium phosphate, pH 7.2 (uncorrected meter reading); in D_2_O; 50 mM KCl, 1 mM TCEP; 1 mM sodium azide and varying levels of DMSO-d6 dependent on ligand solubility; for ligands that were found to be soluble to 200 µM in the above buffer, DMSO-d6 was not used. The STD experiments were carried out using Agilent’s BioPack satxfer1D pulse sequence, with the saturation frequency usually set to ~0.3 ppm and the off-resonance frequency set to 30 ppm. For saturation, 40 × 50 ms Gaussian pulses were applied using a power level corresponding to a Gaussian 90° pulse width of ~0.2–0.3 ppm. In these experiments, the residual water resonance was suppressed using the gradient echo method described by Hwang and Shaka [29].

#### 2.6.2. NMR Assignments

The ^1^H/^13^C-HMQC spectra of the [^13^CH_3_-Met] labeled constructs were acquired using Agilent’s BioPack gChmqc experiment on an Agilent 800 MHz DD2 NMR spectrometer (Agilent Technologies, Santa Clara, CA, USA) equipped with a ^1^H/^13^C/^15^N triple-resonance cryo-probe. The experiments were acquired using WET water suppression [30] applied during the 1 s relaxation delay, with acquisition times of 91.8 ms and 54.1 ms in the direct ^1^H dimension and the indirect ^13^C dimension, respectively. The ^13^C carrier frequency was set to 16 ppm and the ^1^H carrier frequency was set to the residual water resonance.

### 2.7. Computational Ligand Identification

Ligand binding cavities in the p51 subunit of RT that were located at subdomain interfaces were identified based on ligand screening studies reported by Arnold and colleagues [22,23]. Potential ligands included in the database of over 250,000 compounds maintained by the developmental therapeutics program of the National Cancer Institute [31] were screened and ranked using the omega (v. 2.5.1.4) and FRED (OEDocking v. 3.2.0.2) modules in the OpenEye molecular modeling suite (OpenEye, Santa Fe, NM, USA). The OpenEye software package utilizes its own scoring function based on shape complementarity, hydrogen bonding, protein desolvation, and ligand desolvation.

### 2.8. Protein Crystallography/Crystallization

Apo crystals of the FPC1 construct were grown at 4 °C using the hanging drop vapor diffusion technique by mixing 1 µL of 25 mg/mL FPC1 in 10 mM Tris pH 8.0, 50 mM NaCl, and 2 mM DTT with 1 µL of reservoir solution consisting of 0.1 M Tris pH 8.5 and 11% PEG 6000. For data collection, 1 µL of cryo-solution (20% ethylene glycol and 80% reservoir solution) was added to the crystallization drop. The crystal was then transferred to the above-described cryo-solution containing 10 mM of 2,2′,4,4′ tetrahydroxyldiphenyl sulfide, a potential FPC1 ligand, and allowed to soak for 4.5 h. The crystal was then harvested with a cryo-loop and flash frozen in liquid nitrogen for data collection. After solving the structure, there was no density observed for the 2,2′,4,4′ tetrahydroxyldiphenyl sulfide.

Crystals of the FPC1(E203S) construct with picrate were grown at 4 °C using the sitting drop vapor diffusion technique by mixing 270 nL of protein solution consisting of 25 mg/mL FPC1(E203S) in 10 mM Tris pH 7.4, 40 mM NaCl, 1 mM TCEP, 0.25 mM azide, and 10 mM sodium picrate with 270 nL of 0.2 M trisodium citrate and 20% PEG 3350. For data collection, 1 µL of cryo-solution consisting of 15% ethylene glycol and 85% reservoir with 10 mM sodium picrate was added to the well. The crystal was then transferred to the above-described cryo-solution for 30 s prior to flash freezing in liquid nitrogen. We note that, in some crystallization studies of the FPC constructs, we introduced an E203S mutation corresponding to a surface residue on helix F (nomenclature from Wang et al. [32]) near a lattice contact in the hope of facilitating crystallization but, unfortunately, this strategy did not prove useful. However, some of the constructs for which crystals were ultimately obtained contained this additional substitution.

Crystals of the FPC1(E203S) construct with picrate and xanthene present were grown at 4 °C using the sitting drop vapor diffusion by mixing 260 nL of 25 mg/mL FPC1(E203S) in 10 mM Tris pH 7.4, 40 mM NaCl, 1 mM TCEP, 0.25 mM azide, and 10 mM picric acid with 260 nL of reservoir consisting of 0.1 M MES pH 6.5 and 10% PEG 6000. For data collection, the crystal was transferred to a soaking solution of 0.1 M MES pH 6.5, 10% PEG 6000, 10 mM sodium picrate, and 20 mM xanthene-1,3,6,8-tetrol and allowed to soak overnight. The crystal was then transferred to a cryo-solution consisting of 0.1 M MES pH 6.5, 12% PEG 6000, 10 mM picric acid and 20 mM xanthene-1,3,6,8-tetrol, and 20% ethylene glycol prior to flash freezing in liquid nitrogen.

Crystals of the Apo FPC2 construct were grown at 4 °C using the sitting drop vapor diffusion technique by mixing 280 nL of 25 mg/mL protein in 10 mM Tris pH 7.0, 50 mM NaCl, 1 mM TECEP, 5% DMSO, 0.25 mM azide, and 5 mM fisetin with 280 nL of reservoir consisting of 0.1 M TBG at pH 7.0 and 25% PEG 1500. For data collection, 1 µL of cryo-solution consisting of 85% reservoir, 10% ethylene glycol, 5% DMSO, and 5 mM fisetin was added to the drop. The crystal was then transferred to the above-defined cryo-solution for 30 s prior to flash freezing in liquid nitrogen. After solving the structure, there was no density observed for fisetin.

All FPC1 data sets were collected on a Rigaku (Woodlands, TX, USA) MicroMax007HF rotating anode equipped with a Saturn 944 detector, while the FPC2 data set was collected on the same source with a Dectris (Switzerland) Pilatus 200K detector. All data sets were processed using HKL3000 [33]. A modified version of the co-ordinates from pdbcode 4KSE was used for molecular replacement to solve the Apo FPC1 model with the program PHASER [34]. A partially refined model from Apo FPC1 was then used to solve the molecular replacement for Apo FPC2. The Apo FPC2 model was used for MR of the picrate containing FPC1 data and the partially refined model of this was used to solve the FPC1 data with picrate and xanthene present. Structures were refined with iterative cycles of refinement with Phenix and manual model building with Coot [35,36,37,38]. Structure quality was validated with MolProbity [39].

### 2.9. Dimerization Assay by Size Exclusion Chromatography

The effects of various ligands on p51 dimerization were assayed using size exclusion chromatography on an AKTA Pure 25 instrument (GE Healthcare). A Superdex 200 Increase 5/150 column (GE Healthcare) was equilibrated with SEC buffer (50 mM Bis-Tris, 1 M NaCl, 1 mM CDTA, pH 6.5). As noted above, the use of a high salt buffer is important for supporting the formation of p51 homodimer/monomer fractions that can be readily quantified. Samples of 25 µL (80% SEC buffer, 20% glycerol) which consisted of 20 µM p51 protein and a range of different concentrations of potential ligands were injected onto the column and eluted with SEC buffer at a flow rate of 0.8 mL/min. Absorbance at 280 nm was recorded.

### 2.10. SEC Data Analysis

The column selected for the SEC monomer-dimer ratio studies produces overlapping curves corresponding to the p51 monomer and dimer species. Inspection of these traces indicates that the curves are all skewed, tending to rise more sharply than they fall. We initially fit these curves using a skewed normal distribution function available in the program Mathematica (Wolfram Research, Champaign, IL, USA):$$F\left(x\right)=\frac{1}{\sqrt{2\pi }\sigma }e^{-\frac{1}{2}{\left(\frac{x-\mu }{\sigma }\right)}^{2}}\mathrm{Erfc}(\alpha \frac{\mu -x}{\sqrt{2}\sigma })$$
$$F\left(x\right)=\frac{1}{\sqrt{2\pi }\sigma }e^{-\frac{1}{2}{\left(\frac{\left(x-\mu \right)}{\sigma }\right)}^{2}}Erfc\left(\alpha \frac{\left(\mu -x\right)}{\sqrt{2}\sigma }\right)$$
where *µ* corresponds to the position of maximum height (the mode of the distribution), σ to the spread, and *α* is the skew parameter so that values greater than 1 skew curve to the right. Initial fits assumed that the monomer and dimer peaks could be characterized with the same *σ* and *α* values, differing only in *µ*. Mole fraction normalization was achieved by setting 2 × [Dimer] + [Monomer] = 1. However, assuming that the dimer will have approximately twice the absorbance of the monomer, the elution curves are automatically normalized.

## 3. Results

### 3.1. Structural Maturation Pathway and Inhibition Strategy

As originally deduced from molecular modeling studies [32] and subsequently elucidated experimentally [13,40,41], the polymerase domain of p66 exists predominantly in a structurally compact ground state for which the structure has previously been determined (Figure 1A, PDB: 4KSE, [13]); it is similar, although not identical, with the p51 subunit of the RT heterodimer. A very small fraction of the polymerase domain monomer, p51*, adopts an ensemble of distinct structures in which the subdomains have rearranged to form more extended structures similar to that of the p66 subunit in the RT heterodimer. The p51* excited state can form by a unimolecular process that does not require prior dimerization [40]. However, since the p51* structure contains most of the double-stranded nucleotide binding site as well as the NNRTI binding site, the presence of these ligands will increase the fraction of the monomer in the excited state. In order to more clearly emphasize the thermodynamic behavior, we have used the designations ground and excited states rather than the more structurally descriptive compact and extended labels used previously [12,13,40,41,42]. The polymerase domain structural rearrangement is illustrated schematically in Figure 2, left panel, and described by Equation (1), below. At a higher concentration and ionic strength, dimerization of a p51* molecule with a p51g molecule is favored [41], after which some further conformational adjustments supportive of interface formation between the two connection subdomains occur (Figure 2 and Equation (2)). The additional conformational adjustments corresponding to the second equilibration step in Equation (2) arise because subdomain dissociation removes the inter-subdomain interactions, allowing the isolated subdomains to form alternate structures determined only by intra-subdomain interactions and, after initial dimerization, inter-subunit interactions [40,42]. The dimeric structure illustrated in Figure 1C is consistent with the observed polymerase activity of the p51/p51′ homodimer [43,44].



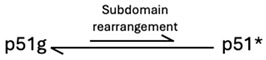

(1)




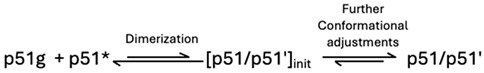

(2)


For p66, formation of the initial p66*/p66g heterodimer similarly involves polymerase subdomain rearrangements and dimerization, followed by formation of an additional interface between the p66* RH domain and the p66g thumb and then by slower transfer of residues from the superfluous p66′ RH′ domain to the p66′ connection’ subdomain, leading to p66′ RH′ domain destabilization, unfolding, and subsequent proteolysis [12,13,14]. These steps occur over a much slower time scale and are ultimately irreversible [45]. Since the initial, reversible maturation steps for p66 and p51 are essentially identical, it is advantageous to investigate the isolated polymerase domain.

The strategy proposed in this study involves the identification of ligands that can bind to the monomer and act as an intramolecular glue that further stabilizes the subdomain interfaces of p51g and inhibits their tendency to dissociate and adopt alternate arrangements. As illustrated on the right-hand side of Figure 2, theoretical inhibitory ligands L1, L2, and L3 that bind to various subdomain interfaces would stabilize these interfaces. Alternatively, inhibitory ligands L4 and L5 would bind to surfaces of the monomer that form a portion of the subunit interface and, hence, compete with dimer formation.

### 3.2. Characterization of Fingers/Palm/Connection (FPC) Constructs

Since the target ligands bind to the more compact, globular fingers/palm/connection subdomain structure, we considered the two disordered regions of the p51 monomer, the thumb and C-terminal segment, to be unnecessary. Such disordered regions of the monomer are difficult to target, and removal of these disordered regions allows us to investigate a globular structure that will, in general, be better behaved, i.e., less likely to undergo proteolytic degradation and more likely to form crystals. In addition, we relied on our previous strategy of deleting a disordered palm subdomain C-terminal segment (Lys219—Met230, referred to as the palm loop or PL) in order to block the metamorphic transition to p51* and, consequently, dimer formation [13,40].

We investigated two constructs designed on the basis of the above considerations. The fingers/palm/connection FPC1 contains the same palm subdomain deletion used in our previous studies of the p51 monomer [13]. The FPC2 construct was based on a recent crystal structure obtained by Bauman et al. as part of a crystallographic screening study for identification of RT ligands and ligand binding sites (PDB: 5CYQ [22]). The disordered palm loop in this structure (Thr215–Lys223) overlaps but differs from the disordered segment in structure 1DLO [9]. More detailed information about these constructs is provided in Appendix A. Since the strategy outlined above involves such major modifications of p51, it was necessary to check that these deletions did not alter the major features of the globular fingers/palm/connection subdomains’ structure. Crystal structures were obtained as described in Methods and crystallographic statistics are summarized in Appendix A.

Figure 3A shows a ribbon diagram overlay of the crystal structure of the FPC1 construct with the p51 subunit (chain B) of unliganded RT (PDB: 1DLO, [9]) and Figure 3B with the isolated p51 monomer (PDB: 4KSE, [13]). As indicated in the figure, the globular fingers, palm, and connection subdomains of FPC1 overlay well with those of both the p51 subunit of RT and the p51 monomer. Alternatively, the unfolded C-terminal residues of the connection subdomain and the deleted thumb are not present in the FPC constructs. Similar structural agreements were obtained using the FPC2 structure. RMSD values corresponding to these comparisons are given in Appendix A.

### 3.3. Initial Ligand Identification

Bauman et al. have performed ligand screening studies on HIV-1 RT that reveal a highly porous surface that is able to accommodate numerous ligands throughout the structure [22,23]. Structural analysis reveals large fissures in the p51 subunit structure that are predominantly located at subdomain boundaries. Due to the structural similarities with the monomer structure, ligands bound to the p51 subunit should also bind to the monomeric forms of p51 and p66 and are potentially useful as maturation inhibitors. The subset of ligands identified by Bauman et al. that interact with p51 while not binding to homologous regions on p66 and that were located at subdomain interfaces was considered to define sites useful for targeting the maturation process. The potential target sites identified based on these studies include the “Glu399 site” located at the F/C interface, which is occupied by a fluorophenyl-methylpyrazole (FMP) ligand: [1-(4-Fluorophenyl)-5-methyl-1H-pyrazol-4-yl]methanol (PDB: 4IFV, [23]), as well as the sites occupied by bromopyrazole molecules 3, 4, 6, and 7 (Figure 4). As apparent from the figure, the FMP ligand in the E399 site corresponds to position L1 in Figure 2, and the various combinations of BYZ ligands bind to positions L1, L2, or L3. The first three molecules bind in close proximity, so that they can be considered to define a single BYZ346 site or smaller BYZ34 and BYZ46 sites. The BYZ7 site is a deep pocket near Met184 that is separate from the BYZ346 site(s). In addition to these ligands, Bauman et al. have also identified a bound sucrose molecule in a region of the F/C interface below the Glu399 site in a study that contained 4% sucrose in the crystallization buffer (PDB: 2I5J, [24]). Nevertheless, NMR transferred NOE studies failed to indicate sucrose binding at 10 mM concentrations, consistent with the absence of significant hydrophobic interactions supporting this interaction. Interestingly, sucrose selects for alternate conformations of Trp24 and Phe61 compared with those observed in the FMP complex. After identification of potential binding cavities in the p51 monomer, the Fred and hybrid modules of the OpenEye Scientific software suite were used to identify and rank optimal ligands from a database derived from the repository of chemical agents maintained by the developmental therapeutics program of the National Cancer Institute [31].

Of several hundred ligands that were identified and ranked by the software, 38 candidate ligands predicted to interact at these sites were obtained in 3–5 mg amounts from NCI Repository of chemical agents maintained as part of the Developmental Therapeutics Program (Appendix A). Structural groups highly represented among the observed hits included flavonoids, phenytoin analogs, substituted diaminopyrimidines, polyhydroxylated biphenyls, and xanthenes. Some additional, structurally related molecules from these groups were obtained from commercial sources and were also evaluated. These ligands were initially screened in groups of three or four using saturation transfer difference (STD) spectroscopy [28]. Most of the ligands yielded only weak responses. The results of two more positive STD screens are shown in Figure 5.

Figure 5A corresponds to a group of NCI ligands that included 9H-xanthene-1,3,6,8- tetrol (XOH) (CAS # 27393-39-1; NCS # 350081), the top-scoring molecule for the BYZ46 site, and the second-ranked compound for binding to the BYZ346 site. The two aromatic XOH proton resonances gave a positive STD signal, while the other high-scoring molecules in the sample showed weak or negligible STD signals (Figure 5A, center spectrum). Another positive STD result was obtained from a second set of ligands that included CAS# 17415-78-0, the third-ranked NCI compound identified to bind to the BYZ346 site by the OpenEye software. As shown in Figure 5B, the STD spectrum showed a single, intense resonance near 9.0 ppm that could not be assigned to any of the compounds used in the STD study. However, the CAS entry CAS# 17415-78-0 indicated that compound N-[3-(2-Amino-6-methyl-4-oxo-1,4-dihydro-5-pyrimidinyl)propyl]-N-(3-aminopropyl)-4-methylbenzenesulfonamide includes a picrate counterion and the 9.0 ppm resonance was found to arise from the picrate. Both picrate and a picrate analog (4-trifuoromethyl,2,6-dinitrophenol) were further evaluated separately and found to produce the strongest STD signals of any of the tested molecules.

The high-scoring OpenEye-identified ligands for the E399 cavity were nearly all azo compounds, which were difficult to evaluate using the chromatographic method described below. We instead evaluated a structurally similar stilbene-4,4′-dicarboxylate compound that appeared similar to several high-ranked ligands such as calmagite.

### 3.4. NMR Studies of [^13^CH_3_-Met]FPC

Although the STD study senses binding, it does not provide information on the location(s) of the bound ligands. Incorporation of [^13^CH_3_]-labeled methionine provides a convenient way of monitoring interactions at multiple protein sites located near these positions [26,41]. The methionine methyl resonances of [^13^CH_3_-Met]FPC1 were assigned using site-directed Met→Leu substitutions, and the assigned resonances are shown in Appendix A. Consistent with previous studies [26], the M16L mutation eliminates two resonances. Based on the methionine methyl ^13^C-chemical shift relationship with the conformational parameter χ^3^ [35], the more intense resonance at δ^13^C = 18.3 ppm likely corresponds to a residue with a predominantly trans χ^3^ conformation [46], as is often observed in the highest-resolution crystal structures, while the weaker resonance at δ^13^C = 17.6 ppm has a greater fraction of gauche χ^3^ conformation. The structural basis for this heterogeneity is unclear. In contrast with previous studies of labeled RT (MW = 117 kD), all of the labeled methionine resonances in the smaller FPC constructs could be observed. Met184 is located near the bound bromopyrazole BYZ7 and, although providing a useful probe for interactions at this site, it was found to produce a broad and weak resonance that is consistent with a relatively dynamic environment. There is unfortunately no Met residue located near the BYZ346 site. In order to obtain some information about potential interactions near this site, we also introduced an L234M mutation into the FPC1 construct. As shown in Appendix A, panel D, introduction of this mutation does not significantly perturb the remaining Met resonances, consistent with the absence of a significant structural perturbation.

The top-scoring flavonoids included a luteolin ester (BYZ346) and robinetin (BYZ7), but additional OpenEye screening indicated that fisetin scored as high or higher for most sites and was somewhat more soluble. Titration of [^13^CH_3_-Met]FPC1(L234M) with fisetin perturbed primarily the M184 and M41 resonances (Figure 6A). The weak M184 resonance was broadened beyond detection after the first addition, demonstrating a perturbation but precluding further analysis of the binding affinity at this site, while a fit of the concentration-dependent shift perturbation of the M41 resonance corresponded to a K_d_ = 1.1 mM (Figure 6B). As noted previously [26], Met16 corresponds to two resonances. The overlapping methyl resonances of Met16b, Met164, and Met234 (present in the L234M mutant) exhibited only small shift changes that did not support specific binding analyses but do not preclude binding at BYZ346. However, the perturbations of the M184 and M41 resonances are consistent with the possibility of fisetin binding near the BYZ7 site as well as involvement of an additional site near Met41. Chemical shift perturbation patterns similar to those produced by fisetin (Figure 6A) were observed for several other ligands, including the top-scoring BYZ46 ligand, 9H-Xanthene-1,3,6,8-tetrol (XOH). The ambiguities of interpreting these results were largely resolved by crystallographic studies described below.

Interaction of fisetin with the FPC construct was also evaluated using CB-Dock software, v. 1 [47], described in greater detail below. This analysis indicated that fisetin may interact with several of the target sites in p51, including sites near Met184 and E399. In the latter, the chromone ring of fisetin stacks against Trp24 similar to the methylpyrazole ring of the FMP ligand (PDB: 4IFV, [23]). These results are consistent with the NMR titration data summarized above and with the effects of fisetin in the p51 dimerization assay described below.

### 3.5. Crystallographic Characterization of Ligand Binding

Although crystal structures were obtained for both FPC1 and FPC2, the FPC constructs did not crystallize readily, perhaps due to the dynamic behavior of several loops [42,48]. However, picrate, which gave a strong STD signal (Figure 5B), was also found to promote crystallization. The FPC–picrate complex contains a cluster of three picrate molecules located in a surface depression of the fingers subdomain between the β2–β3 and β7–β8 loops (Figure 7). The three negative charges are neutralized by residues Lys20, Lys22, and Arg143, with the latter forming a stacking interaction with picrate 1 (Figure 7A). The β7–β8 loop, which has been shown to play an important role in forming the interface with p66 [49,50,51], follows a different trajectory in the presence of the picrate ligands. The well-defined picrate cluster was observed in two different crystal structures (Appendix A) and defines a specific arrangement of the component picrate molecules. Omit maps for the bound picrate and xanthene ligands are shown in Appendix A. The picrate cluster also forms a lattice contact at which two FPC molecules interact, with the PIC3 molecule common to both clusters (Appendix A). Lys20 and Lys20′ interact with the common PIC3 ligand, while Lys22, forms long intermolecular H-bonds with picrate molecules 2 and 3, as well as a shorter H-bond with PIC1′ that is bound to a symmetry mate. Facilitation of crystallization is supported by these direct intermolecular contacts and probably by the stabilization of specific conformations of the two flexible loops indicated above.

Consistent with β7–β8 loop flexibility, the positions adopted in crystal structures of the p51 subunit of the RT heterodimer, the p51 monomer, the FPC construct, and the FPC–picrate complex all differ. Overlay of the FPC–picrate complex with the p51 subunit of RT indicates direct steric conflict with p66 residues Trp88 and Glu89 on the p66 αB-β5 loop (Figure 7B). Additionally, the picrate 2 position conflicts with p51 residue Asn138. Thus, both direct steric competition with p66 and alteration of the positions of the β2–β3 and particularly the β7–β8 loop in p51 will interfere with dimerization.

Due to the greater ease of obtaining FPC1–picrate crystals, FPC1 was crystallized in the presence of both XOH and picrate ligands. In the predicted complex, XOH overlays the BYZ4 and BYZ6 ligands and is sandwiched between palm subdomain Y231 and connection subdomain Q373 and W410 residues, with the hydroxyl groups making multiple H-bonds (Figure 8A). Nevertheless, XOH was not present in the BYZ46 site predicted by the modeling but instead bound to a site near BYZ7 and to a second site near Met41 (Figure 8B,C). In the bound structures, the Met184 sidechain at site BYZ7 adopts a conformation that differs from that observed in the apo structures or in 5CYQ, creating a binding pocket, so that XOH1 is sandwiched by Met184 on one side and the peptide backbone and Lys154 on the other. Binding selectivity is supported by H-bond interactions with Asn81 and Asp185 (Figure 8B). These interactions are consistent with the high affinity of the Met sidechain for aromatic groups [52].

The Met41 site containing bound XOH2 is formed from a depression in the fingers subdomain and is not located at a subdomain interface. In addition, a structural comparison of the Met41 ligand binding site in the two subunits of the RT heterodimer indicates that they are quite similar. It is, therefore, unlikely that this interaction will significantly influence dimerization or structural maturation.

### 3.6. Comparison with a Previously Identified Dimerization Inhibitor

Liu et al. [47] have recently developed a blind docking protocol (CB-Dock) that identifies protein cavities ranked by size and positions a user-identified ligand in each of these cavities, providing a Vina score to quantify the binding interaction. In general, the nature of these cavities is dependent on the positions of the protein sidechains, so that applications of this software generally yield more useful results using structures obtained in the presence of ligands that are then removed in silico prior to the docking calculation. CB-dock calculations performed on the isolated p51 subunit of RT structure 5CYQ or using the structures of the FPC constructs identified most of the ligand cavities described above but failed to identify the cavity occupied by the picrate ligands. However, the picrate binding cavity was identified by CB-dock if the protein structure determined in the presence of picrate was used. After completion of the above studies, we used the blind docking approach to evaluate interactions with several other ligands that have been reported to interfere with dimerization. Docking of the N-acyl hydrazine dimerization inhibitor (BBNH) identified by Sluis-Cremer and colleagues [51] in the FPC construct with the picrate ligands removed reveals it to exhibit significant affinity for the picrate cavity. The overlay shown in Figure 9 indicates that the BBNH 4-t-butylbenzoyl ring occupies the position of PIC1 and the BBNH naphthyl ring occupies the position of PIC2. The FPC1-bound BBNH conformation is stabilized by specific interactions involving the linking segment that connects the two rings. The unprotonated nitrogen forms a H-bond with Gln23 NH, and the linking carbonyl oxygen forms H-bonds with Thr131 and with the Asn57 amide nitrogen. Docking studies using BBNH analogs containing various modifications of the linking segment indicate that most modifications significantly impaired the fit predicted for the resulting ligand. These results indicate that (1) part of the ability of BBNH to interfere with dimerization may involve interaction with the fingers binding cavity observed in the presence of picrate and (2) the identification of this potential BBNH binding site may allow further iteration of the BBNH structure to improve affinity.

### 3.7. SEC Analysis of p51 Dimerization

Dimerization was followed both by NMR and by size exclusion chromatography (SEC); however, the former approach is not optimal due to low sensitivity and the long accumulation times required. Although the p66 monomer and dimer can be readily separated by size exclusion chromatography, separation of p51 monomer and dimer is more challenging, primarily because the dimer fraction is generally quite low. However, the p51 dimer fraction can be significantly stabilized at high ionic strength [41], similar to the behavior of p66 [53,54]. The effects of ionic strength on the p51 homodimer/monomer ratio were readily verified chromatographically (Figure 10A) and are consistent with previous studies using other methods [41]. As expected, the dimer/monomer ratio also increases as a function of p51 concentration (Figure 10B) and the data were quantified based on the simulations shown in Figure 10C. Previous analyses of p66 and p51 homodimerization have been based on simple monomer–dimer equilibria that require only a single dissociation constant, K_d_, or using a model similar to that described above, in which dimerization is dependent on two equilibrium constants, K_d_ and K_m_, that describe the dissociation and metamorphic equilibria, respectively [41]. However, the experimental SEC data indicating that some dimer is observed at very low concentrations, while the concentration-dependent increase exhibits a low slope that is not well described by either of these models. As illustrated in Figure 10D, the data can be fit using either of the above models but constraining the limiting dimer fraction to a value dependent on the ionic strength and other experimental conditions. The limiting dimer fraction of 0.47 used for the analysis in Figure 10D supports a model in which, under the conditions of the study, the energetic cost of forming the excited state p51* is close to the energetic gain resulting from dimer formation (Appendix A). At lower ionic strength, the free energy gain resulting from dimerization will be even less (Appendix A), so that, even at a very high concentration, the limiting dimer fraction will be even lower, consistent with the results of Figure 10. Alternatively, addition of efavirenz (EFV), an NNRTI shown to support heterodimer formation [55,56], substantially increases the p51 dimer/monomer ratio, as indicated by quenching of the p51 intrinsic fluorescence (Appendix A). Similar quenching of p66 + p51 intrinsic fluorescence is observed upon heterodimer formation [57]. Dimer stabilization by efavirenz results primarily from its selective binding to and stabilization of either p66* or p51*, which contain the NNRTI binding site [14].

Although preliminary studies showed that a number of the ligands tested reduced the dimer fraction as anticipated, the effects generally required millimolar levels. Results obtained for the bioflavonoid fisetin and for picrate, which directly interferes with dimer formation (Figure 7), are shown in Figure 11. These studies also encountered a significant experimental problem resulting from protein precipitation, which typically was not immediate but occurred over varying time periods after addition of the ligand. It is straightforward to understand this effect for ligands such as picrate that directly interfere with dimer formation. In this case, the ligand directly interferes with stabilization of the p51* by blocking dimer formation. The p51* conformations are intrinsically unstable—indeed, one of the primary functions of dimerization is stabilization of the active p66 polymerase fold [42]. Protein precipitation is more difficult to understand for ligands that target buried p51 sites. Perhaps interactions of these ligands with specific sites in the monomer may interfere locally with the metamorphic transition, leading to a greater tendency to form partially but incompletely rearranged structures that are less stable and have a greater tendency for further unfolding or aggregation and precipitation. For the SEC assay, we found that addition of 20% glycerol prevented or greatly limited this precipitation while not interfering with the dimerization assay or altering the monomer/dimer fraction.

The ligand-dependent SEC data were not easily fit to a simple, single-site reversible binding model. In the case of picrate, the concentration-dependent formation of a ternary picrate cluster observed in the crystal suggests why this might be the case. Additionally, formation of the picrate–p51 complex involves selection of specific conformations for the β2–β3 and β7–β8 loops, resulting in an entropic penalty. We instead determined the ligand concentrations at which the fractional dimer intensity was reduced to half its value in the absence of ligand. IC50 values for fisetin and picrate were determined to be 2.2 mM and 5.2 mM, respectively. An IC50 = 7.6 mM was obtained for cynaroside, a high-scoring BYZ346 ligand noted above (Appendix A). We also evaluated stilbene-4,4′-dicarboxylate as a structural analog of calmagite, a high-scoring E399 cavity ligand. In the predicted structure, the calmagite sulfonate group forms a salt bridge with Arg78 and a stilbene carboxylate would be similarly positioned. A preliminary SEC study gave an IC50 = 6 mM. Disappointingly, the xanthene tetrol was found to act as a weak inhibitor, with an IC50 > 10 mM. As noted above, binding to the Met41 site is unlikely to influence the maturation process, while binding near the BYZ7 site apparently has only a weak effect.

## 4. Discussion

### 4.1. Subdomain Interface Binding Sites and Ligand Identification

As recently noted in a review by Ruiz and Arnold, greater understanding of RT maturation opens the door to unprecedented opportunities for therapeutic intervention [58]. In order to form a mature RT heterodimer, a pair of initially equivalent p66 proteins must develop along distinct pathways, resulting from formation of an asymmetric homodimer [59]. Early structure-based analyses led to predictions of asymmetric homodimer structures by Steitz and colleagues [32] that have been supported and extended, primarily based on NMR studies of Ile-labeled p66 and its isolated subdomains [12,13,14,40,41,42]. We previously demonstrated that a short deletion near the C-terminus of the palm subdomain could block the essential metamorphic transition, allowing observation of a stable monomer. The goal of the present studies was the identification of ligands targeting the subdomain interfaces of the p66 monomer that would stabilize and support subdomain interfaces in the ground state and thereby block or inhibit the required metamorphic transition. Since the polymerase domains in p51M and p66M adopt essentially identical structures [13] (Figure 1), the initial steps of p66 and p51 maturation involve similar structural transformations (Figure 2), allowing us to investigate these steps using the isolated polymerase domain (p51). This has the advantage of eliminating the slower and functionally irreversible steps related to RNase H domain unfolding and proteolysis.

Two different fingers/palm/connection (FPC) subdomain constructs containing the same interfaces that are present in the monomers and in the p51 subunit of the RT heterodimer were investigated using both NMR spectroscopy and X-ray crystallography. It is remarkable that the basic structural features of this construct can be maintained despite major sequence deletions (Figure 3 and Appendix A). The subdomain interfaces identified in previous ligand screening studies [22,23] are relatively polar, consistent with the spontaneous subdomain dissociation and rearrangements required as part of the maturation process (Figure 4). The broad M184 resonance is also consistent with dynamic behavior in this region of the monomer. These characteristics makes the identification of high-affinity ligands challenging, since druggability is predominantly driven by pocket hydrophobicity [60,61]. Effective targeting of these regions thus requires very specific ligands in order to recognize the complex surfaces of these sites. NMR studies of a ^13^CH_3_-methionine-labeled FPC construct revealed that several ligands, including the bioflavonoid fisetin (Figure 6) and the high-scoring ligand 9H Xanthene-1,3,6,8-tetrol, interact near Met41 and Met184, and crystallographic studies with XOH confirmed that it also binds near these residues (Figure 8). The M41 site is not located at a subdomain or a dimerization interface and the p66 and p51 structures in the region of M41 overlay closely. The M184 site is potentially useful for further development of maturation inhibitors; however, as noted above, the effect of XOH on p51 homodimerization was extremely weak. Overlay of the FPC-XOH complex with the p51 subunit of RT bound to bromopyrazole ligands shows that BYZ7 is located near the bound XOH (Figure 8B). It may be possible to use this information in order to develop a ligand that occupies both the BYZ7 and XOH sites, leading to enhanced contacts with both the palm and connection subdomains.

### 4.2. Direct Interference with Dimerization

Since the dimeric nature of RT was first identified, extensive efforts have been made toward the development of dimerization inhibitors [50,51,62,63,64,65,66,67]. In most cases, these studies investigated heterodimer formation by p66 and p51, although there is strong evidence supporting initial formation of a p66/p66′ asymmetric homodimer followed by subunit-selective proteolysis to form the heterodimer [13,68]. Additionally, some reported dimerization studies investigated heterodimer formation kinetics from an initial preparation of p66 and p51 monomers stabilized in a buffer containing ~20% acetonitrile. However, information on the structure of the monomers present under these conditions is limited, and Venezia et al. [69] have reported that the association-kinetic behavior determined in standard buffers is considerably slower, consistent with the requirement for a metamorphic transition prior to dimerization [40]. We emphasize that, for both heterodimer and homodimer formation, the starting monomer structures exist primarily as p66g or p51g (Figure 1), either of which needs to undergo a metamorphic transition to p66* or p51* prior to dimer formation. To further emphasize this point, p66M, which exists primarily as p66g, needs to form p66* prior to dimerization with p66g to form an asymmetric homodimer.

Since the subunit interface of the RT heterodimer is not directly accessible to solvent, experimental studies designed to probe interactions of ligands with these interfaces cannot be performed. Alternatively, the stabilized p51 monomer construct (PDB: 4KSE, [13]) is well suited for such evaluations, since this includes a portion of the heterodimer interface that is not occluded by the p66 subunit. However, we are unaware of studies using the stabilized monomer to identify dimerization inhibitors. The FPC constructs described in the present study may be more useful for such measurements, since these lack most of the disordered components that generally limit crystallization efforts. STD studies performed using the FPC construct identified picrate as a high-affinity ligand, and X-ray crystallography revealed a trio of picrate molecules bound to the fingers subdomain surface involved in dimer interface formation. Picrate was also demonstrated by size exclusion chromatography to interfere with p51 homodimer formation (Figure 11). This result illustrates the potential of even an untargeted ligand search using a construct, in which a region of the protein inaccessible to solvent in the homo- and heterodimer structures is solvent-accessible and, thus, ligand-accessible in the monomer structure.

The effects of picrate on FPC crystallization are also worth noting. Picrate was observed to interact with cationic Lys and Arg residues on the protein surface that often are mobile or disordered and can interfere with crystallization. The picrate complex not only formed a more stable structure but also a lattice contact in which Lys residues were found to make intermolecular as well as intramolecular contacts (Appendix A). Similar picrate locations at lattice contacts combined with picrate-mediated intermolecular interactions appear to be present in a few other structures, e.g., N-acetylornithine aminotransferase (PDB: 4JEW, 4JEX, Bisht et al., 2013 doi.org/10.2210/pdb4JEX/pdb). These observations suggest that picrate may be a more generally useful additive for protein crystallization studies. Unexpectedly, the loop conformations stabilized by picrate binding also were sufficient to define a binding cavity that was more readily recognized by the CB-dock software [47] and then used to dock additional ligands, including the previously identified dimerization inhibitor BBNH (Figure 9), providing a basis for further structural iterations or ligand identification.

## 5. Conclusions

HIV-1 reverse transcriptase achieves an economy of genome size by utilizing a complex structural maturation pathway requiring rearrangement of its polymerase subdomains. In this study, we have explored the possibility of interfering with RT structural maturation by identifying ligands that can target the subdomain interfaces, inhibiting their ability to dissociate and reassociate. Since the ground state of the p66 monomer is structurally similar to the p51 subunit of the RT heterodimer, p51 subunit binding ligands previously identified in screening studies can serve to identify binding cavities that have been used for computational screening of ligand libraries. In this study, a preliminary set of computationally identified ligands from a database of chemical agents maintained by the Developmental Therapeutics Program of the National Cancer Institute were evaluated as inhibitors of the initial steps in the structural maturation pathway. Although the results reported here support the feasibility of this approach, the initially identified ligands produce only modest effects, and this limited efficacy appears to be related to the dynamic and structural characteristics of the targeted binding cavities. In addition to screening more extensive ligand libraries, further evaluations of other classes of compounds such as peptides may prove to be a useful strategy for targeting these complex binding sites. This study also demonstrates the utility of using the stabilized polymerase monomer in screening for dimerization inhibitors that directly target the subunit interface that is inaccessible in the RT heterodimer.

## Figures and Tables

**Figure 1 biomolecules-13-01603-f001:**
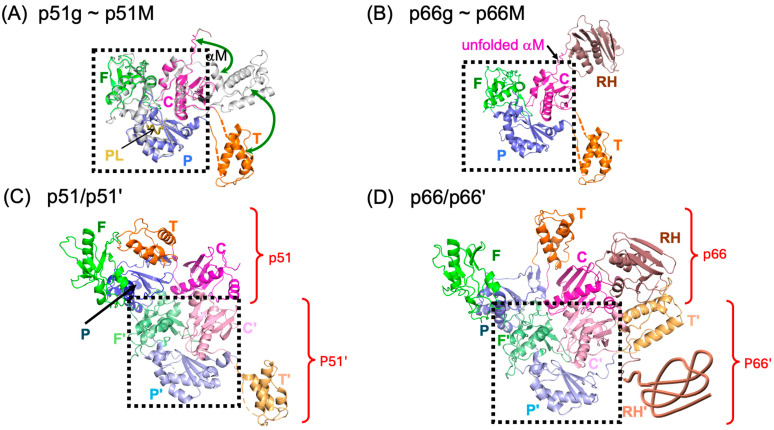
Structures of p51 and p66 monomers and homodimers. (**A**) Ribbon diagram of the stabilized p51 monomer lacking the disordered palm loop residues (PDB: 4KSE, [1]) in which the position of the missing palm loop residues is indicated (PL, gold loop). The subdomains are color-coded: fingers (F, green), palm (P, blue), thumb (T, orange), and connection (C, magenta), and the structure is shown overlaid with the p51 subunit of RT (gray, PDB: 1DLO, [2]) in order to illustrate the differences. (**B**) The p66 monomer structure determined based on NMR studies of the δ-^13^CH_3_-Ile-labeled constructs [13]. The RH domain is shown in brown. The monomer structures in panels A and B correspond to the predominant p51g and p66g ground states. (**C**) The modeled p51/p51′ homodimer is derived from the structure of the p66/p51 heterodimer by deleting the p66 RH domain and replacing the p51 subunit structure with that of the p51M shown in panel A. Since p51 lacks an RNase H domain, there are no inter-subunit contacts to position the T′ subdomain in p51′. The compact, globular FPC structure indicated by the dotted black line is present in each of the structures shown. (**D**) The p66/p66′ homodimer is formed after a series of slow processes that include formation of the thumb’–RH interface. Late in the process, residues are transferred from the folded RH′ domain to the C′ subdomain. The p66/p66′ homodimer shown is based on RT heterodimer structure 1RTJ [3], in which the entire RH′ domain segment from Tyr427 through Phe440 has been transferred to C′. In this state, the RH′ domain has already unfolded but, in the illustrated structure, RH′ has not yet been proteolyzed and is represented as a random coil. In the dimeric structures, panels C and D, the active subunits are shown using brighter hues and the structural subunits are shown using paler hues. The helical segment at the C-terminus of the C′ subdomain extends through Leu29, incorporating residues that also are important for RH′ domain stability.

**Figure 2 biomolecules-13-01603-f002:**
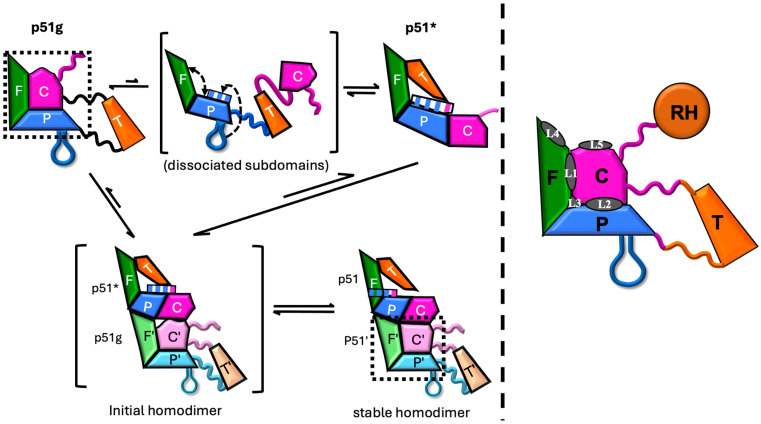
Reversible dimerization of the RT polymerase domain and identification of inhibitory targets. (Left hand panel): rearrangement of the p51 monomer proceeds through dissociation of the polymerase subdomains with the exception of the discontinuous fingers/palm. The dissociated subdomains can either reassemble back to the more stable ground state or rearrange and alternately reassociate to adopt an ensemble of more extended p51* structures that approximate the active polymerase structure in the p66 subunit of the heterodimer. The palm loop (blue) is able to form a short β-sheet, adding an additional strand from the connection subdomain (magenta) on the inner surface of the palm subdomain. As shown in the lower portion of the figure, the p51* excited states can then form an initial dimer with the monomer that is further stabilized by conformational adjustments of the palm and connection subdomains. Deletion of the palm loop (blue) prevents subdomain rearrangements and formation of p51* [13]. Subdomain color coding: fingers (green), palm (blue), thumb (orange), and connection (magenta), with the p51′ subdomains of the homodimer indicated with paler hues. The more compact, globular FPC structure characteristic of the p51 subunit is also present in the initial monomer ground state and is indicated by a dotted black line. (Right hand side): schematic of p66g showing potential subdomain and interface target ligands postulated to stabilize the monomer form. Ligands L1, L2, and L3 would target subdomain interfaces, while ligands L4 and L5 would target the p66g monomer surface involved in dimer formation. Color coding is as in the left-hand panel, with the additional RH domain shown in brown.

**Figure 3 biomolecules-13-01603-f003:**
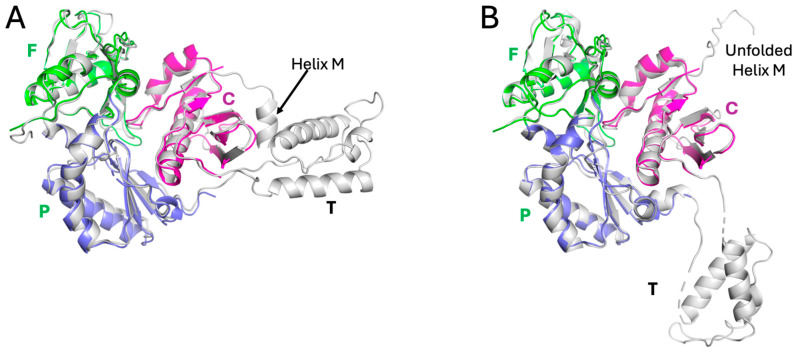
Structural comparisons of FPC constructs. (**A**) Ribbon diagram overlay of construct FPC1 with the p51 subunit (chain B) of unliganded RT (gray, PDB: 1DLO, [9]). (**B**) Overlay of FPC1 with the stabilized p51∆PL construct lacking palm loop residues Lys219-Met230. This palm loop deletion previously was shown to stabilize the compact subunit structure of the p51 monomer [13], and the same palm loop residues are disordered in the p51 subunit of apo RT (PDB: 1DLO). The FPC1 subdomains are color-coded as in Figure 1. As also shown in panel B, the p51∆PL segments linking the thumb to the palm and connection subdomains are partially disordered and helix M has unfolded.

**Figure 4 biomolecules-13-01603-f004:**
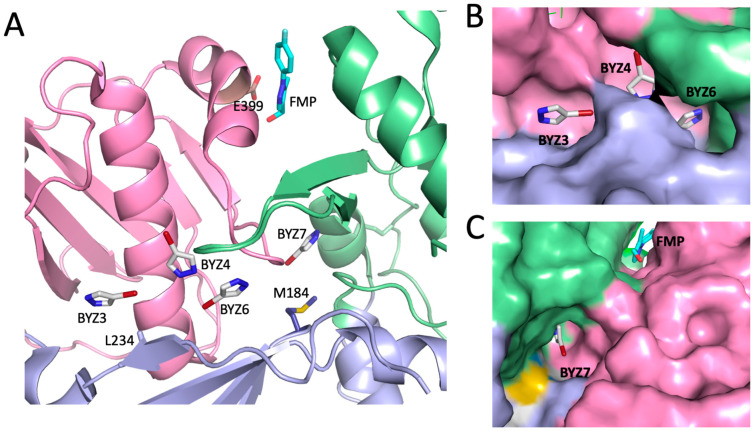
Positions of previously identified ligands located at subdomain boundaries. (**A**) The positions of bromopyrazoles BYZ3, BYZ4, BYZ6, and BYZ7 and of the Glu399 ligand [1-(4-Fluorophenyl)-5-methyl-1H-pyrazol-4-yl]methanol (FMP) in the p51 subunit of RT were obtained using overlaid PDB files 5CYQ [22] and 4IFV [23]. The protein is shown as a ribbon diagram and the sidechains of Met184 located near BYZ7 and L234, located near BYZ3 and E399, and located near FMP are also indicated. (**B**) Surface rendering of the FPC1 binding region containing BYZ3, BYZ4, and BYZ6; (**C**) surface rendering of the FPC1 binding region containing BYZ7 and the Glu399 ligand. In all panels, the subdomains are color-coded as fingers (green), palm (blue), and connection (magenta).

**Figure 5 biomolecules-13-01603-f005:**
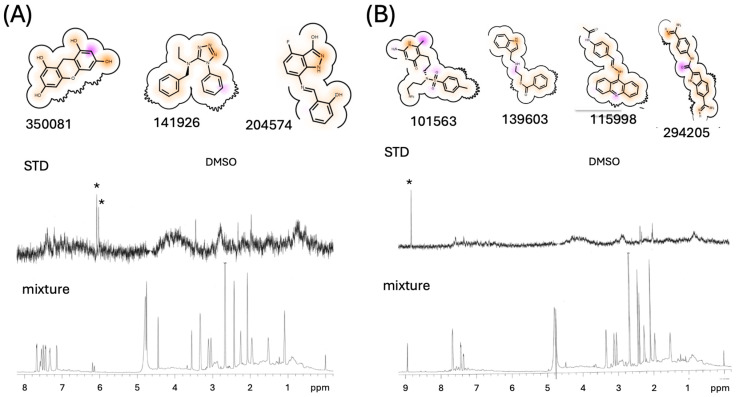
Illustrative STD data. Panels (**A**,**B**) correspond to screening studies of the ligands pictured at the top. The lower spectra are control NMR studies of the mixture, and the spectra in the center of the panels are the STD experiments. In panel A, compound nsc350081 (tetrahydroxyxanthene) produced significantly stronger STD peaks (indicated with an asterisk) compared with the other compounds present in the mixture. In panel B, only the resonance near 9.0 ppm gave a strong STD signal (indicated with an asterisk). As noted in the text, this resonance was assigned to the picrate counter ion present in nsc101563 (CAS # 17415-78-0). The figures at the top include the compound structures present in the group, the NSC number, and a schematic representation of the binding site; positions highlighted in yellow make favorable H-bonds with the binding site, while positions highlighted in magenta make unfavorable contacts. The STD samples contained 200 µM of the test compounds in the following screening buffer: 15 µM FPC1 in 50 mM K_2_HPO_4_, pH 7.4; 50 mM KCl; and 10% DMSO-d6 in D_2_O containing a DSS shift standard.

**Figure 6 biomolecules-13-01603-f006:**
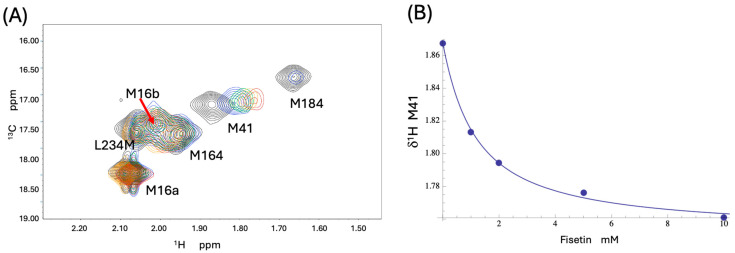
Effect of fisetin on the NMR spectrum of [^13^CH_3_-Met]FPC1(L234M). (**A**) Titration of methionine-labeled FPC1 with fisetin produces significant shift perturbations of Met41, while Met184 shifts and is strongly broadened, precluding a more complete titration analysis. The remaining Met resonances exhibit smaller shifts, which are more difficult to interpret quantitatively due to overlap. As shown in Appendix A, M16 produces two resonances. (**B**) The interaction with Met41 corresponds to a K_d_ = 1.1 mM. The sample contained 200 µM of the labeled FPC1 in 50 mM KH_2_PO_4_ (pH7.5), 50 mM KCl, 20% DMSO-d6, 80% D_2_O, where the DMSO was required to maintain fisetin solubility over the concentration range used. The sample was in a Shigemi tube and the experiment run on a Varian 800 MHz NMR spectrometer at 25 °C.

**Figure 7 biomolecules-13-01603-f007:**
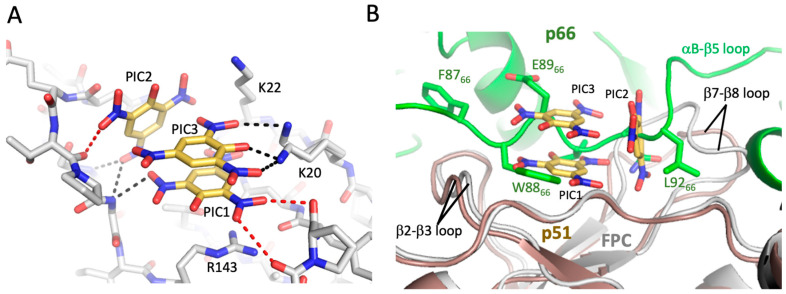
Picrate binding to the fingers subdomain interferes with dimerization. (**A**) Three picrate ligands (gold) are bound to the fingers subdomain in the crystal structure with FPC1 (gray). The three negative charges are neutralized by Lys20, Lys22, and Arg143 in the binding site. The β2–β3 loop shifts position and the conformation of the β7–β8 loop is altered relative to their positions in the unliganded FPC structures. In the FPC1–picrate complex structure, the picrate ligands also form a lattice contact with a second FPC1 molecule (Appendix A). As indicated in the figure, the electron density of Lys20 was fit to two alternate conformations. Dashed black lines indicate H-bonds and dashed red lines indicate repulsive interactions. (**B**) Overlay of the picrate–FPC1 complex with the p51 subunit of apo RT (p66 (green)/p51 (brown); PDB: 1DLO, [9]) demonstrates significant steric conflict with the p66 αB-β5 palm loop. In the overlay, the position of picrate 1 largely overlaps the Trp88 indole ring in p66 subunit of the RT heterodimer, with picrate 2 and 3 showing extensive conflict with other residues in the p66 αB-β5 palm loop. In addition to the direct overlap, alteration of the conformation of the p51 β7–β8 loop would interfere with p66 binding. Thus, picrate binding to p51 would be expected to compete with formation of the p66-p51 dimer and with formation of the p66-p66′ homodimer.

**Figure 8 biomolecules-13-01603-f008:**
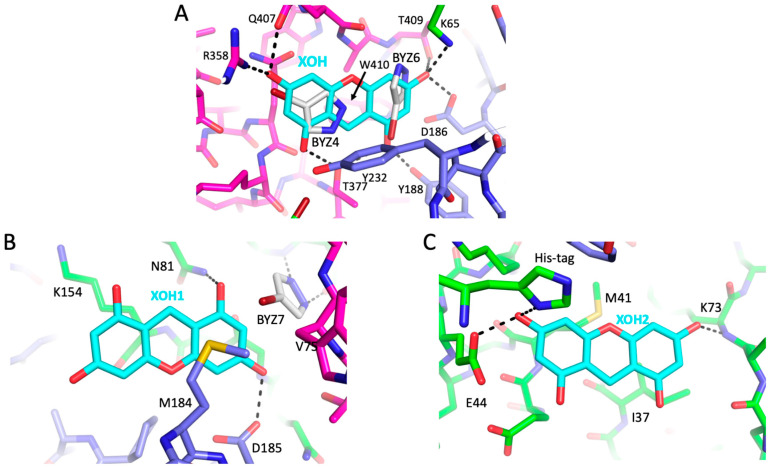
Predicted and experimental interactions with XOH. (**A**) Predicted position of 9H-Xanthene-1,3,6,8-tetrol (XOH, cyan) in the BYZ46 site of FPC1. The ligand is sandwiched between palm subdomain residue Y232 and connection subdomain residues Q373 and W410, with the hydroxyl groups positioned to form multiple H-bond interactions. Predicted hydrogen bonds with Lys65 and Arg358 sidechains are not likely to be significant since their positions vary in different crystal structures. (**B**) Observed interactions of the XOH1 ligand at the BYZ7 site. The XOH hydroxyl groups form H-bonds with the Asp185 and Asn81 sidechains and the xanthene ring is stacked between Met184 and Lys154. (**C**) Binding of XOH2 at the Met41 site in the fingers subdomain; the XOH hydroxyls form H-bonds with the Glu44 sidechain, the Lys73 backbone amide, and with a His-tag-derived imidazole. The ring stacks against Ile37 and is positioned near Met41. Subdomains are color-coded as in Figure 1. The shift behavior of Met41 in the presence of fisetin (Figure 6) suggests that it also is binding to this site.

**Figure 9 biomolecules-13-01603-f009:**
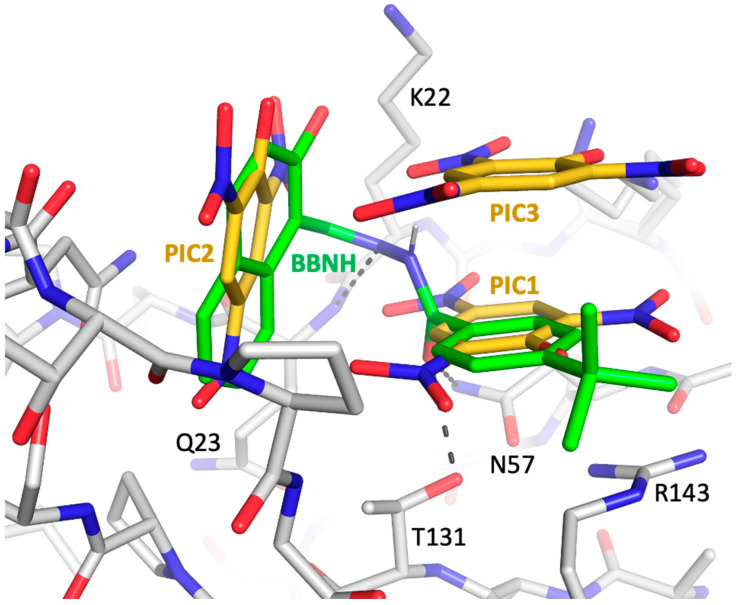
CB-dock analysis showing binding of dimerization inhibitor BBNH to the picrate site. A CB-dock analysis of the interaction of ligand BBNH (green) with the FPC1 construct (gray) was performed on the structure of the complex after removal of the bound picrate (gold) and tetrahydroxyxanthene ligands. The docking program is unable to identify this binding cavity in structures obtained in the absence of picrate. The BBNH aromatic groups overlap with the PIC1 and PIC2 ligands (shown after superimposing the picrate ligands on the BBNH complex) and the linking segment makes H-bonds with Gln23, Asn57, and Thr131, as indicated. As in Figure 7, density for K20 is resolved into two alternate conformers.

**Figure 10 biomolecules-13-01603-f010:**
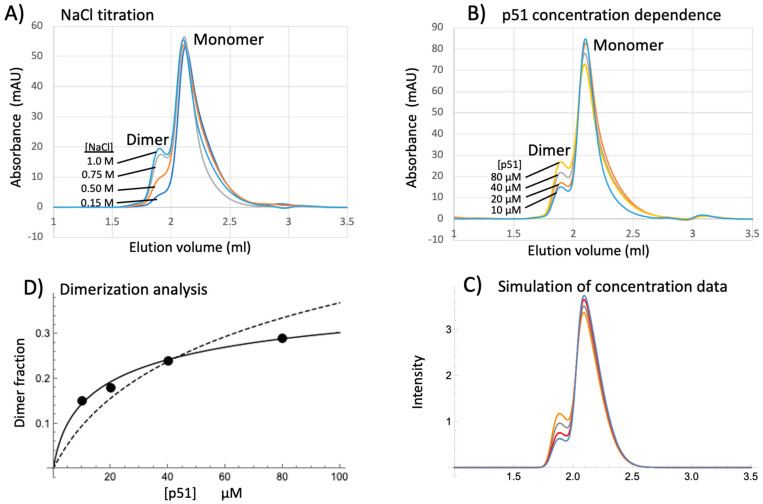
Size exclusion chromatograms showing the effects of ionic strength and concentration on p51 homodimerization. (**A**) The homodimer fraction increases as a function of [NaCl] = 0.15, 0.5, 0.75, 1.0 M. (**B**) Homodimer fraction increase for [p51] = 10, 20, 40, 80 µM. (**C**) Simulation of the data in panel B using a skewed Gaussian model with parameters µ1 = 1.82, µ2 = 2.03, σ1 = σ2 = 0.17, and α1 = α2 = 4.5. The orange, gray, red, and blue colors used in panels B and C both correspond to the concentrations indicated in panel B. (**D**) Fit of the homodimerization data (filled circles) using a model in which the limiting fractional homodimer approaches 1.0 (dashed line) or a limit of 0.47 (solid line); the comparison indicates that only the latter model provides a reasonable fit of the data.

**Figure 11 biomolecules-13-01603-f011:**
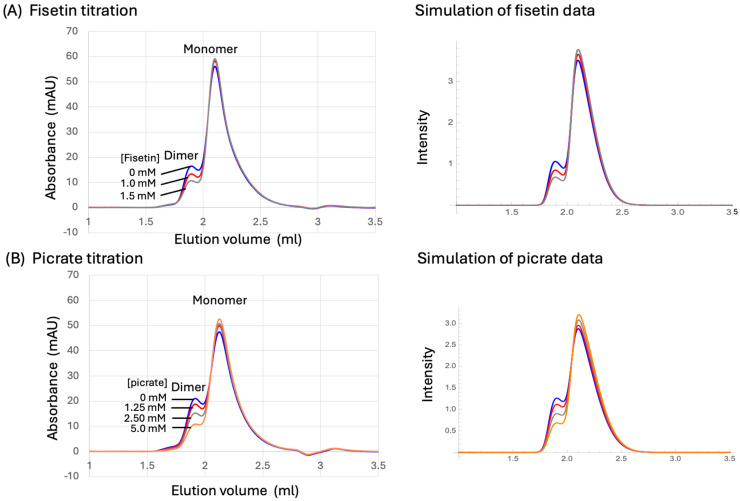
Effects of fisetin and picrate on p51 homodimerization. (**A**) Size exclusion chromatogram of 20 µM p51 as a function of fisetin (0, 1.0, and 1.5 mM); (**B**) size exclusion chromatogram of 20 µM p51 as a function of picrate (0, 1.25, 2.5, and 5.0 mM). A set of simulated curves obtained as described in the text is shown to the right of each set of titration data. Colors used for the simulated curves on the right correspond to the color-concentration relation indicated on the left. The fractional dimer concentration decreases by 50% at 2.2 mM fisetin or 5.2 mM picrate. The SEC column was equilibrated and eluted with SEC buffer (50 mM Bis-Tris, 1 M NaCl, 1 mM CDTA, pH 6.5); samples contained 80% SEC buffer, 20% glycerol.

## Data Availability

Crystallographic data has been deposited in the protein data bank (www.rcsb.org) under PDB accession numbers: 8TCK (apo FPC1), 8TCL (FPC1-picrate complex), 8TCM (FPC1-picrate-Xanthene tetrol complex) and 8TCJ (apo FPC2). Additional information concerning these depositions is contained in Appendix A.

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
