# Peer review of "Targeting the Structural Maturation Pathway of HIV-1 Reverse Transcriptase"

_biomolecules, 2023, doi:10.3390/biom13111603_

Round 1
Reviewer 1 Report
Comments and Suggestions for Authors
The authors obtain four complexes of FPC protein; they check binding of previously identified inhibitors, and estimate the binding of a new small molecule library. A selection of best molecules were experimentally evaluated, using also methods to check the ligand binding cavity. The aim is to target a specific conformation of the protein that undergoes to a conformational change during his maturation.
The manuscript is well organized; all the work is well described and method section is detailed, figures complement the text.
Author Response
We appreciate the time spent in looking at our manuscript. No specific responses have been made in response to reviewer 1.
Reviewer 2 Report
Comments and Suggestions for Authors
The article “Targeting the structural maturation pathway of HIV-1 reverse transcriptase” by Kirby et. al. is well written. However, I have few minor comments.
1- Line 177-178 – “The crystal was then transferred to 100% cryo-solution” – its not clear what does it mean by 100 % cryo-solution. It needs to be clarified.
2- In data statistic table (table 1) – some more relevant parameters should be included like Rmeas, Rpim, CC1/2, CCwork, CCfree.
3- There are no density maps shown for any of the ligands. A figure showing omit map around the ligands of interest should be included with the manuscript.
4- There are some grammatical mistakes/typos in the manuscript like line 48 (maturation process of required to form the), line 194 (contained included) etc.
Comments on the Quality of English LanguageExcept few minor typos, english language is good.
Reviewer 3 Report
Comments and Suggestions for Authors
In this manuscript, Kirby et al report on the structural and biochemical characteristics of deletion mutants of the p51 subunit of HIV-1 reverse transcriptase (RT) in an effort to understand the molecular mechanisms involved in the formation of the RT p66/p51 heterodimer. Authors have obtained p51 polypeptides with deleted palm subdomains and studied the dimerization equilibrium of p51 in the present of picrate or fisetin that impide the formation of p51 dimers.
Authors have collected quite a lot of information but the paper is written in an unattractive manner. The flow of information is not clear. The results section contains a lot of discussion, particularly in subsection 3.1. This is even recognized by the authors that state in p. 8 (line 336) that “As discussed more fully below…”. Results section should contain only results and previous evidence should be presented in succinct manner, and only the necessary facts to understand the motivation of the study and the major findings. Even the introduction is a better place to present some of the data discussed in section 3.1.
Another problem related to data presentation is that some of the supplementary figures are important to understand the paper, most notably, Suppl. Figure 2 that is cited ahead of Suppl Figure 1 in the paper (p. 3, line 100). The sequence alignments should be included in the paper. The structure of FPC1 and FPC2 is not obvious to the reader. Also Suppl Figure 1 contains relevant structures that in my opinion are more relevant than the working model shown in Figure 1. This model is poorly explained in the legend. In contrast Tables 1 and 2 are more suitable for the supplementary section.
On the technical side, there are several issues that need more attention:
(1) There is little clarity about how compounds have been selected, ranked and studied. In Fig. 4, authors show six compounds that appear to be p51 binders, but only NSC350081 and NSC101563 are nominated and discussed. There are no given names for the formulae and description of these results is difficult to follow.
(2) Picrate seems to adopt different poses when binding p51. Is this related to an expected unspecificity? Considering the results given in Figure 10, these are very weak inhibitors and their significance in dimerization inhibition is questionable.
(3) Authors dedicate attention to 9H-xanthene-1,3,6,8-tetrol (XOH). The molecule has been identified as a binder in the initial screening and Figure 7 shows experimental interactions with p51. However, I am missing some evidence suggesting an effect of this compound in the dimerization equilibrium (Fig. 10).
(4) This paper lacks a punchline, I do not see a clear conclusion of the study. It reads like a collection of data lacking an integrated interpretation. I feel that the article needs to be rewritten and better organized.
(5) Although p51/p51 is considered to be inactive, some activity could be probably detected in DNA polymerization assays in the presence of Mn2+. It would be interesting to assess the inhibitory activity of the molecules discussed in the paper and see how data correlate with the reported findings.
Minor issues:
.- p. 1, lines 38-39: Cleavage of p66 to obtain p51 involves the removal of 120 amino acids of the C-terminal of p66, remaining 440 amino acids in p51. Obviously it is not half of the molecule.
.- p. 6, Legend to Fig. 1, line 267: “inhibitory”
.- p. 8, line 340. The structure of compounds I4 and I5 should be presented somewhere in the manuscript.
Comments on the Quality of English LanguageEnglish grammar and usage are fine. Only minor corrections needed. However, the structure of the paper and the presentation of the results is poor and in my opinion, requires extensive re-thinking and re-writing.
Round 2
Reviewer 3 Report
Comments and Suggestions for Authors
Authors have improved the readability of their manuscript, although the paper is still difficult to understand, particularly for non-specialists. The study is technically complex and assumptions are not obvious and are sometimes based on predictions.
Some issues need further clarification:
(1) At the beginning of the Results section (p. 7) authors refer to p51g, p51*, p66g. However, there are no explanations about the structures indicated by using these acronyms. Why “g”? In the legend to Fig. 2, what are the specific characteristics of p66g? I guess that authors represent p66g in the right hand scheme, but unlike in p51g, here we can see inhibitors bound to the structure. This needs further clarification.
(2) Figure 2 shows in the left lower part an initial homodimer that is in equilibrium with a stable homodimer. What kind of forces are expected to contribute to this stabilization? Please, explain.
(3) In line 450 (page 11), authors refer to E99 cavity but I guess they mean E399.
(4) Page 1, lines 38-41: Although the transformation of p66/p66 to p66/p51 can be observed in vitro, It has not been demonstrated whether the cleavage of the RNase H domain occurs in vivo, before or after the formation of p66/p66.
(5) Authors have identified a number of compounds that are potential monomer binders. In the paper, I miss two important points: (i) IC50 estimates or accurate binding affinities for each of them; and (ii) how these compounds relate to I1 to I5 in their mechanism of action. It is possible that in some cases, there are no data or the mechanistic interpretation would be difficult, but at least this should be mentioned as a limitation of the study.
